# Numerical Study of Thermal Enhancement in a Single- and Double-Layer Microchannel Heat Sink with Different Ribs

**DOI:** 10.3390/mi13111821

**Published:** 2022-10-25

**Authors:** Miaolong Cao, Shi Cao, Jincheng Zhao, Jiayi Zhu

**Affiliations:** School of Mechanical and Energy Engineering, Zhejiang University of Science & Technology, Hangzhou 310023, China

**Keywords:** single- and double-Layer microchannel heat sink, heat transfer enhancement, numerical simulation

## Abstract

In this paper, a microchannel heat sink model was proposed to realize single- and double- layer flow through built-in ribs. The finite element volume method was used to analyze the influence of the length, thickness and angle of the inner rib on the flow and heat transfer characteristics of the microchannel heat sink. The pressure drop, temperature field, flow field, and thermal characteristics are given. The numerical simulation results show that the rectangular rib plate makes the fluid in the microchannel heat sink flow alternately in the upper and lower layers, which can effectively enhance heat transfer. However, with the increase in rib length, the comprehensive evaluation factor decreases. The change of the angle of the rectangular rib plate has little influence on the Nusselt number. The change rate of the comprehensive evaluation factor of the thickness of the rectangular rib plate is the largest. When the Reynolds number is 1724, the comprehensive evaluation factor of Case 9 is 4.7% higher than that of Case 2. According to the parameter study of the built-in rib plate, the optimal parameter combination is given, in which the angle is 0°, the length is 7.5 mm, and the thickness is 0.2–0.3 mm.

## 1. Introduction

With the continuous upgrading of modern electronic equipment, the size and scale are getting smaller and smaller, and the integration is getting higher and higher, resulting in higher heat flux density and a shorter service life. Therefore, the problem of heat dissipation of equipment has always been a research hotspot in the field of heat transfer [1,2]. As early as 1981, Tuckerman and Pease [3] studied microchannel heat sink for the first time and used purified water as the cooling medium, which successfully dissipated the heat with a heat flux of 1000 W/cm^2^. The microchannel heat sink has the advantages of large specific surface area, simple structure, small mass, small volume, and high convective heat transfer coefficient [4,5]. Lee and Garimella [6] studied a rectangular microchannel with a width range of 194–534 μm and a depth of five times the width. The effectiveness of the classical correlation to predict the single-phase flow thermal behavior of rectangular microchannels is shown. The results show that the numerical prediction is in good agreement with the experimental data, and the average error is 5%. The traditional Navier Stokes equation is still applicable. Husain et al. [7] used the response surface method to optimize the trapezoidal groove structure. Kumar [8] compared the heat transfer characteristics of the trapezoidal microchannel heat sink and rectangular microchannel heat sink at a Reynolds number of 96–720. The results showed that the heat transfer of the trapezoidal groove was 12% higher than that of rectangular microchannel heat sink. At the same time, by adding rectangular grooves and circular grooves to the trapezoidal microchannel heat sink, they can further improve the heat transfer performance. Chai et al. [9] conducted experimental research on the laminar flow and heat transfer of water in two trapezoidal silicon microchannels. The pressure loss of the import and export buffers in the microchannel will be smaller than that of no buffers. Under the given pump power condition, the thermal resistance will decrease with the heat flux.

With the progress of CFD (computational fluid dynamics) technology, the numerical simulation can maintain good efficiency while solving problems, and the calculation results are relatively reliable. Therefore, the performance of microchannel heat sinks is widely used and studied [10]. Tokgoz [11] carried out experimental and Numerical Research on the flow characteristics in the pipe with circular corrugated channels on the wall. The comparative data show that the velocity distribution changes periodically, and the error between simulation results and experimental results is small. Wang [12] compared the cooling effects of ribbed microchannels and traditional smooth rectangular microchannels by experimental and numerical methods. The results indicate that in the same condition, the Nusselt number of rib-grooved microchannels can be 1.11–1.55 times that of the smooth microchannel. Through systematic experiments, Yang [13] measured two square channels whose opposite surfaces were roughened by ribs with a high blockage rate. The heat transfer coefficient of symmetrically arranged ribs is higher than that of staggering ribs, but the pressure loss of symmetrically arranged ribs is greater than that of staggering ribs.

To improve the heat transfer performance of the microchannel, related researchers chose to optimize the microchannel structure, such as built-in ribs of different shapes [14,15], and double/multi-layer microchannel heat sinks [16,17]. Xie et al. [18] designed five kinds of built-in ribs with different lengths in the rectangular microchannel heat sink, The results show that, compared to the smooth microchannel heat sink, the existence of built-in ribs can reduce the thermal resistance, and the length of built-in ribs can effectively improve the heat transfer performance. Leng et al. [19] proposed an improved top truncated double-layer microchannel heat sink. The proper cutoff design of the top channel can reduce the heating effect of the top while maintaining the fluid cooling effect. Han [20] proposed a microchannel heat sink with built-in rib plate, which can realize the flow separation in the microchannel. The numerical simulation results show that when the fluid passes through the vertical bifurcation, there is an obvious inflection point in the pressure gradient, and the microchannel heat sink with short distance between the end of the internal vertical bifurcation and the outlet of the microchannel has the best thermal performance. Cao [21] proposed a novel multi baffle type heat sink, which effectively improved the maldistribution of temperature and had good heat transfer performance. Wang [22] designed a v-rib microchannel heat sink using water-based nanofluid as coolant. The results show that the periodic arrangement of the v-rib can improve the convective heat transfer performance. Chai [23] designed five offset ribs with different shapes. The offset rib has a significant heat transfer enhancement effect and large pressure drop, but the advantage of the offset rib microchannel heat sink is that is an effective heat transfer enhancement method that gradually loses at a high Reynolds number. Derakhshanpour K. R. [24] designed a new type of single and double-layer microchannel. The results show that the addition of cylindrical ribs can lead to the redevelopment of chaotic advection and a thermal boundary layer, and then affect the heat transfer performance. Zhang et al. [25] has designed a structure of two layers of fins with different thickness, which can effectively guide more cryogenic fluid to the lower layer, reduce pressure drop loss and promote high heat transfer.

According to previous research by scholars, microchannels with built-in ribs and multi-layer flow can improve the heat transfer performance of a microchannel’s heat sink, but the two aspects have not been combined. Therefore, a new model is proposed in this paper to achieve single- and double-layer flow through built-in ribs. The length, thickness and angle of the built-in rib are important parameters that affect the overall thermal and flow characteristics of the microchannel. For this reason, the numerical heat transfer performance of the built-in ribbed plates with different lengths, thicknesses and angles is compared to that of the traditional smooth channels, and the reasons are analyzed.

## 2. Description of Physical Models

As shown in Figure 1a, the overall model of the microchannel heat sink consists of a length of 50 mm and a width of 22.5 mm, with a total of 10 channels. At present, the multi-channel with symmetry is adopted [26]. In the simulation, to reduce the grid node division and calculation time, the researchers used single channel calculation, as shown in Figure 1b. The detailed geometric parameters of microchannel heat sinks are as follows: The cover plate is 0.5 mm, the total width of single channel is 3 mm, the channel height is 1 mm, the channel width is 2 mm, and the height from the bottom of the microchannel to the bottom of the fluid is 0.5 mm.

Based on the traditional smooth microchannel (Case 0), a series of microchannel radiators with built-in ribs are designed. Before the fluid encounters in the first built-in rib, it maintains a single-layer flow, while when it encounters the built-in rib, the fluid is forced to be divided into two layers and kept a distance. The fluid returns to a single-layer flow again before passing through the first internal rib and encountering the second internal rib. Since the number of built-in ribs is two, the flow of fluid in the channel is double-layer flow and single-layer flow is three times.

The geometric parameters of the built-in rib plate are shown in Table 1, where L is the length of the rectangular rib plate and d is the thickness of the rectangular rib plate, θ is the angle of the rectangular rib plate, and the designed structural model is shown in Figure 1. Water is selected as the cooling working medium. In actual work, the relevant parameters of water will change with the temperature [17]. The material of microchannel is aluminum [27], and the density of aluminum: ρ = 2719 kg/m^3^, specific heat capacity: $C_{p}$ = 871 J/kg∙K, thermal conductivity: $\lambda_{s}$ = 202.4 W/m∙K.

## 3. Computational Method and Procedure

### 3.1. Mathematical Model

This study was carried out using a three-dimensional simulation, and some assumptions [28] were made. They are as follows:Three-dimensional, steady-state, incompressible, laminar flow.Ignore gravity and thermal radiation heat transfer.Adopt inviscid dissipation

Therefore, the governing equation of fluid can be described as:

Mass equation
(1)$$\frac{\partial }{\partial x_{i}}\left(\rho_{f}u_{i}\right)=0$$

Momentum equation
(2)$$\;\;\;\frac{\partial }{\partial x_{i}}\left(\rho_{f}u_{i}u_{j}\right)=-\frac{\partial p}{\partial x_{j}}+\frac{\partial }{\partial x_{i}}\left[\mu_{f}\left(\frac{\partial u_{j}}{\partial x_{i}}+\frac{\partial u_{i}}{\partial x_{j}}\right)\right]$$
where i,j=1,2,3

Energy equation
(3)$$\frac{\partial }{\partial x_{i}}\left(\rho_{f}u_{i}c_{f}T\right)=\frac{\partial }{\partial x_{i}}\left(\lambda_{f}\frac{\partial T}{\partial x_{i}}\right)$$

In which $\;x_{i}$, $x_{j}$ are cartesian coordinate components, $u_{i}$ and $u_{j}$ are velocity components, p is the pressure in the basin, $\rho_{f}$ is the density of fluid, T is the temperature, $\mu_{f}$ is the dynamic viscosity of the fluid, $c_{f}$ is the specific heat capacity of fluid, and $\rho_{f}$ is the density of the fluid. $\lambda_{f}$ is the thermal conductivity of the fluid.

For solids, only heat conduction is considered.
(4)$$\frac{\partial }{\partial x_{i}}\left(\lambda_{s}\frac{\partial T_{s}}{\partial x_{i}}\right)=0$$
where $\lambda_{s}$ is the thermal conductivity of the solid region and $T_{s}$ is the temperature of the solid region.

Reynolds number at the inlet is defined as:(5)$$Re=\frac{\rho vD_{h}}{\mu }$$
where ρ is the fluid density, $u_{in}$ is the average velocity of the fluid at the inlet of the channel, $D_{h}$ is the hydraulic diameter, and μ is the dynamic viscosity.

Nusselt number (Nu) represents the heat transfer performance caused by convection and is written as
(6)$$Nu=\frac{hD_{h}}{\lambda_{f}}$$

The average heat transfer coefficient [29] can be expressed as:(7)$$h=\frac{Q}{A_{w}\Delta T_{m}}$$
where $Q=q.A_{b}$, $A_{b}$ is the area of the bottom surface, and q is the wall heat flux. $A_{w}$ is the heat transfer area. The mean temperature difference $\Delta T_{m}$ is between the wall and the water.

Friction coefficient is defined as:(8)$$f=\frac{2\Delta pD_{h}}{L\rho_{f}u_{in}^{2}}$$
where Δp is the pressure drop and $u_{\mathrm{in}}$ is the inlet velocity.

The total thermal resistance of the heat sink is calculated as:(9)$$R_{th}=\frac{T_{max}-T_{\mathrm{in}\;}}{Q}$$
where $T_{\max}$ is the maximum substrate temperature and $T_{\mathrm{in}\;}$ is the inlet temperature.

### 3.2. Boundary Conditions

Based on the finite volume method, ANSYS Fluent 2020R^1^ was used to simulate the following cases. A uniform heat flux of 100 W/cm^2^ is applied on the bottom surface of the substrate. The boundary conditions of the inlet and outlet in the fluid computational domain are set as velocity-inlet and pressure-outlet, respectively, and the initial temperature of the fluid in the microchannel is 300 K. The uniform inlet velocity varies from 0.5 to 1.5 m/s. A no-slip boundary condition is applied to the solid-liquid coupling surface. The wall of the microchannel is set as an adiabatic boundary.

In this study, the governing equations are solved via the second-order upwind method and the pressure-velocity coupling via the SIMPLE method. Standard discretization is applied for pressure field. When the residual values of them are less than 10^−6^ for all variables, the numerical solutions are considered to be converged.

### 3.3. Mesh Independence and Model Validation

To improve the calculation accuracy and calculation time during simulation, the effectiveness of the grid is verified. The overall structure adopts unstructured grid division, which is densified near the built-in rib plate. The size of the grid element is from 0.01 to 0.02 mm. The grid division results are shown in the Figure 2. When the inlet velocity is 1.5 m/s, the grid generation results of three groups of Case 1 are compared. The results are shown in Table 2. Mesh III is regarded as the baseline. The deviation of Nu for Mesh II is 0.18%, respectively, and that of Mesh I is 2.96%, respectively. The results show that the number of grids has little effect on the simulation. Therefore, Mesh II meets the simulation requirements.

### 3.4. Experiment and Simulation Verification

Figure 3 shows the experimental equipment diagram. When the water temperature is 300 K, the peristaltic pump will deliver the water to the microchannel radiator through the pipeline. The flow range of the peristaltic pump is 1–100 mL/min. The inlet and outlet pressures are measured by pressure gauges, and the range is 0 to 0.2 Mpa. The accuracy class is 2.5. The pressure drop of microchannel radiator is the difference between inlet and outlet. Place a heating device under the aluminium plate. Meet the constant heat flow boundary of *q* = 100 W/cm^2^. The heating plate is in close contact with the bottom of the microchannel radiator, and the heat flow is transferred from the heating plate to the microchannel.

As shown in Figure 4a, the experimental data and simulation data of pressure drop in rectangular microchannels are compared in the velocity range of 0.5–1.5 m/s. The error of simulation and experiment data is about 10%. The error source may be the influence of external temperature, pipeline pressure loss, etc., but these errors are within a reasonable range. Similarly, the experimental data on the maximum temperature at the bottom of the paper by Ho et al. [30] are also verified, as shown in Figure 4b. The results show that, although there are errors between the simulation data and the experimental data, the simulation data still has some guiding significance.

## 4. Results and Discussion

### 4.1. Influence of Rib Length

Figure 5 shows the pressure drop distribution of Cases 1–4 at different inlet velocities. With the increase in inlet velocity, the pressure drop of four groups of microchannel heat sinks also increases. At the same inlet velocity, the increase in the length of the rectangular rib in the microchannel heat sink will increase the pressure drop in the channel. This is because, when the length of the rectangular rib plate increases, the fluid encounters the rib plate, the velocity increases, and the pressure potential energy is converted into kinetic energy.

As shown in Figure 6, when the inlet velocity is 1.5 m/s, the streamline diagram of Case 0–4 shows that the fluid flow in the channel is mainly controlled by viscous force due to the small number of built-in ribs and low velocity. Even if the presence of built-in ribs increases the fluid disturbance in the channel, the streamline still appears smooth.

As shown in Figure 7, when the velocity is 1.5 m/s, the velocity cloud at the first rectangular rib of Cases 1–4. The rectangular rib with different lengths has a certain impact on the velocity inside the microchannel heat sink. Before the entrance section and the rectangular rib appear, the velocity remains stable. With the appearance of rectangular rib, the region where the fluid passes is compressed, forcing the fluid to pass at a higher velocity, and establishing a new boundary layer. The change of velocity affects the surrounding pressure, thus enhancing heat transfer. When the fluid leaves the rib, the velocity begins to return to the original state.

As shown in Figure 8, when the inlet velocity is 1.5 m/s, the temperature distribution cloud diagram at the bottom of Cases 1–4. When the fluid passes through the rectangular rib plate, the temperature drops obviously. However, after leaving the first rib plate and before reaching the second rib plate, the temperature will rise locally. Similarly, when the fluid flows through the second rib, the temperature will decrease again, but it cannot reach the minimum temperature. The long-distance of the rib plate increases the secondary heat transfer between the solid and the fluid. Therefore, the lowest temperature (307 K) appears in Case 4, which indicates that the presence of ribs reduces the temperature of the bottom plate. Figure 9 shows the temperature distribution along the mainstream direction at the centerline of the bottom microchannel heat sink. The range of temperature changes in Case 4 is the gentlest. Compared to Case 0 and Case 1, the minimum temperature has decreased by 1.95% and 1.3%, respectively, which will help the uniform distribution of the bottom plate temperature.

As shown in Figure 10, when the inlet velocity is 1.5 m/s, the temperature map of the solid area in Case 1–4 shows that the existence of rectangular rib enhances the disturbance of fluid, thus damaging the thermal boundary layer, improving the heat transfer, thereby reducing the temperature of the heat transfer surface. Due to the influence of the inlet effect, the length of the rib greatly reduces the temperature near the rib.

The comparison of average Nusselt numbers of models 1–4 at different Reynolds numbers is shown in Figure 11. The average Nusselt number of the four groups of microchannel heat sinks will increase with the increase in the Reynolds number, which indicates that Cases 1–4 can effectively improve the heat transfer characteristics. Among them, the average Nusselt number of Case 3 is the largest. When the Reynolds number is equal to 1989, compared to Case 0, the average Nusselt number of Case 3 is increased by 29.9%. It is fully explained that the length of the rectangular rib plate will affect the heat transfer characteristics of the microchannel heat sink, but the length of the rib plate needs a suitable value.

As shown in Figure 12, the thermal resistance of different models decreases with the increase in the Reynolds number. The results show that the thermal resistance of microchannel heat sink with rectangular ribs is lower than that of smooth microchannel heat sink at the same Reynolds number. Additionally, the difference in thermal resistance between microchannel heat sinks with different lengths is small.

### 4.2. Influence of Rib Thickness and Angle

Figure 13 shows the pressure drop distribution under Cases 5–9 under different inlet velocities. With the increase in inlet velocity, the pressure drop of five groups of microchannel heat sinks also increases. The difference of their pressure drop is small between Cases 5–7, which indicates that the angle change of the built-in rib plate has little effect on the pressure drop. At the same Reynolds number, the pressure drop increase rate caused by the thickness of the built-in rib plate is the largest. When the Reynolds number is 1989, the pressure drop of Case 9 is increased by 83.9%, as compared to Case 5, and the pressure drop of Case 8 is increased by 43.2%.

As shown in Figure 14a, The Nusselt number of Cases 5–9 is shown to change with Reynolds number. With the increase in the Reynolds number, the Nusselt number of each Case increases gradually. Under the same Reynolds, the change of Nusselt number caused by the thickness of the built-in rib is far greater than the angle change of the built-in rib, and the difference between different angles of the built-in rib is small. When the Reynolds number is 1459, the Nusselt number of Case 9 is increased by 25.1% compared with Case 5. Figure 14b shows the change of thermal resistance of Cases 5–9 at different Reynolds numbers. With the increase of Reynolds number, the thermal resistance gradually decreases. At the same Reynolds number, the thermal resistance of the thickness of the built-in rib plate is lower than the angle of the built-in rib plate. This is because, compared with the volume increase caused by the angle change of the built-in rib plate, the volume increase caused by the increase of the thickness is larger, and the heat transfer region between the fluid and the built-in rib plate can be almost ignored. This shows that the role of the volume of the built-in rib plate on the thermal resistance has a negative correlation, and increasing the volume of the built-in rib plate can effectively reduce the thermal resistance.

### 4.3. Overall Thermal Performance

To analyze the influence of the length, angle, and thickness of the rectangular rib on the overall heat transfer performance and pressure drop of the microchannel, a comprehensive evaluation factor, Nu/Nu_0_/(ΔP/ΔP_0_)^1/3^, was introduced, where (ΔP/ΔP_0_)^1/3^ directly represents the increase in pressure drop, while Nu/Nu_0_ represents the improvement of convective heat performance. The comprehensive evaluation factors are all greater than 1, which indicates that Cases can effectively enhance in thermal performance [31,32]. The results are shown in Figure 15. As far as the length of the built-in rib plate is concerned, the comprehensive evaluation factor first increases and then decreases with the length of the built-in rib plate, and in Case 2, the comprehensive evaluation factor is the largest. As far as the angle of built-in rib plate is concerned, the comprehensive evaluation factor decreases gradually with the increase in angle. Compared to the angle of the built-in rib plate, the difference of comprehensive evaluation factors caused by the length change of the built-in rib plate is small, and the heat transfer performance is better than the angle change of the built-in rib plate. With the increase of the thickness of the built-in rib plate, the change rate of the comprehensive evaluation factor gradually increases. When the Reynolds number is small, the comprehensive evaluation factor decreases with the increase in the thickness of the built-in rib plate, but with the gradual increase of the Reynolds number, the comprehensive evaluation factor increases with the increase in the thickness of the built-in rib plate. This is because the Cases with built-in rib thickness needs greater pump power to show better heat transfer performance.

## 5. Conclusions

In this study, the numerical simulation of three-dimensional flow and heat transfer in rectangular microchannels with rectangular ribs of different lengths, angles, and thicknesses is carried out by using computational fluid dynamics. The flow and heat transfer characteristics of smooth microchannels were compared. The conclusions are as follows:The existence of built-in ribs makes the fluid in the microchannel alternate between single and double layers, which can effectively enhance the heat transfer of the rectangular microchannel heat sinks.In four groups of rectangular ribbed plates with different lengths, the Nusselt number of Case 3 is 29.9% higher than that of Case 0, and the thermal resistance is 23.6% lower. The comprehensive evaluation coefficient of Case 2 is the highest, 3.57% higher than that of Case 4.Among the four groups of rectangular rib plates with different angles, the change of angle has little influence, but in terms of thickness, the change rate of the comprehensive evaluation factor is the largest, and the pump power is also the largest.

## Figures and Tables

**Figure 1 micromachines-13-01821-f001:**
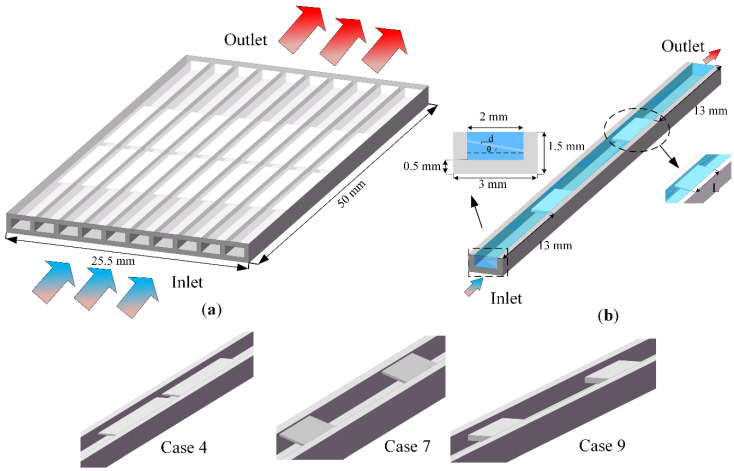
Schematic diagram of microchannel heat sink. (**a**) Overall microchannel structure; (**b**) microchannel heat sink in the computational domain.

**Figure 2 micromachines-13-01821-f002:**
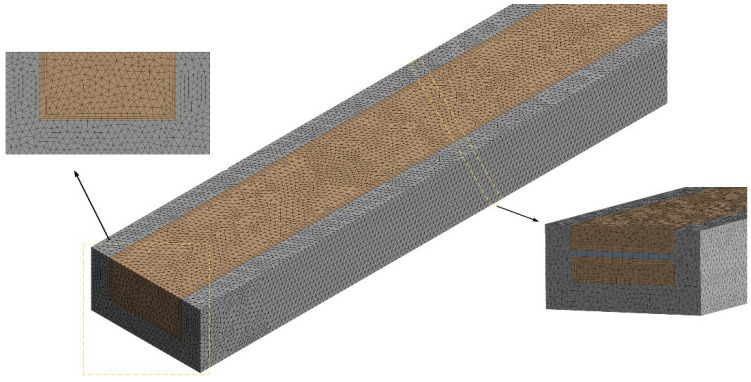
Schematic diagram of grid generation results.

**Figure 3 micromachines-13-01821-f003:**
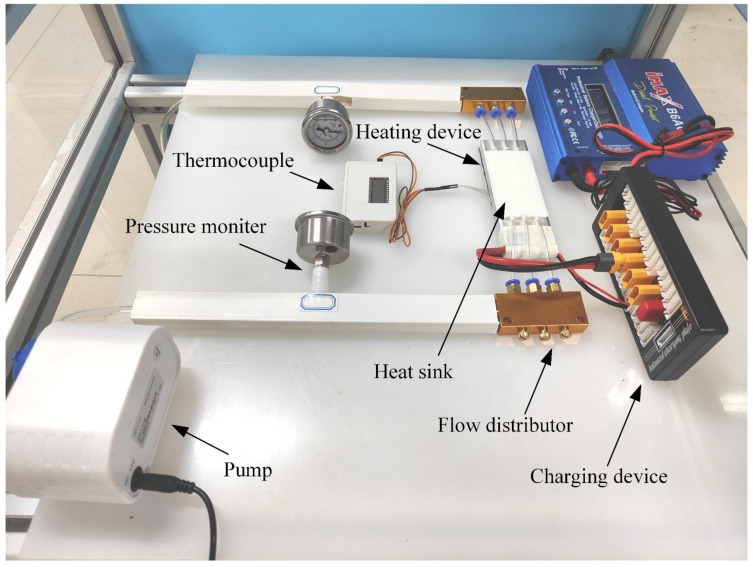
Experimental equipment diagram.

**Figure 4 micromachines-13-01821-f004:**
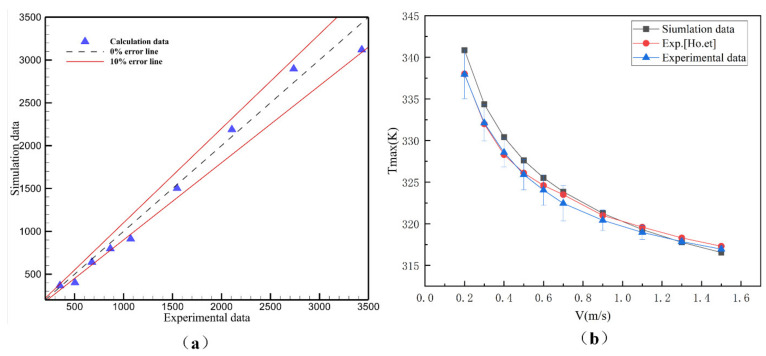
(**a**) Comparison diagram of experiment and simulation of pressure drop; (**b**) comparison diagram of bottom maximum temperature experiment and simulation.

**Figure 5 micromachines-13-01821-f005:**
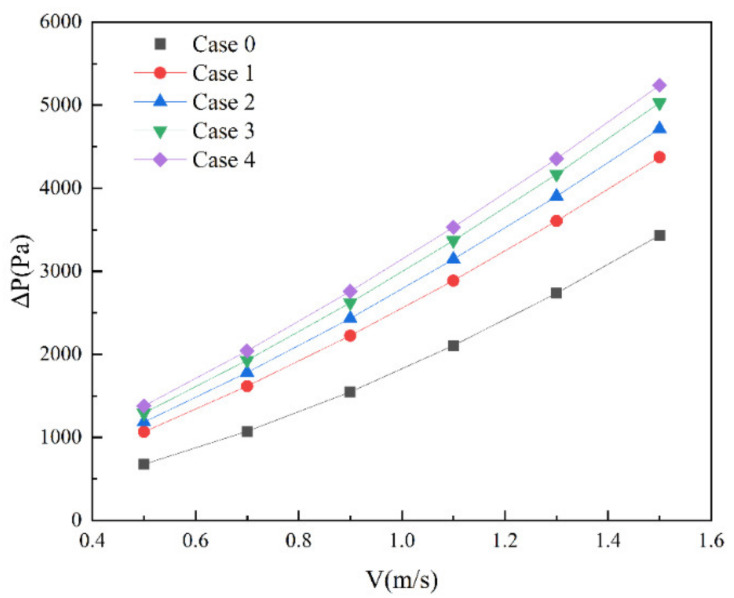
Pressure drop of Cases 0–4 at different inlet velocity.

**Figure 6 micromachines-13-01821-f006:**
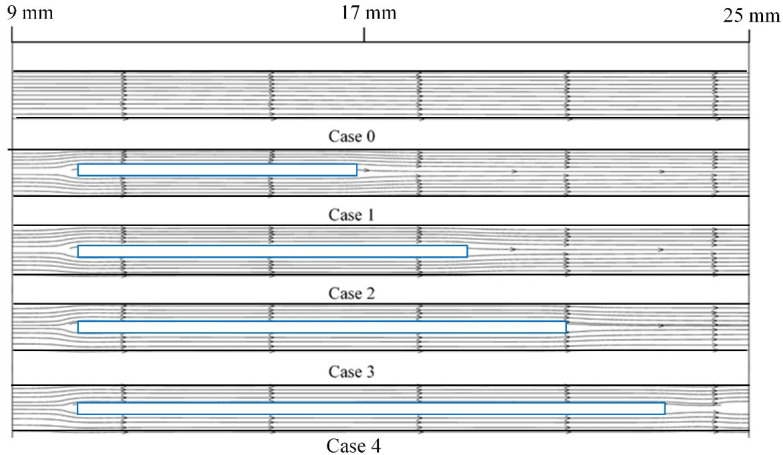
Streamline diagram of Case 0–4 at the inlet velocity of 1.5 m/s.

**Figure 7 micromachines-13-01821-f007:**
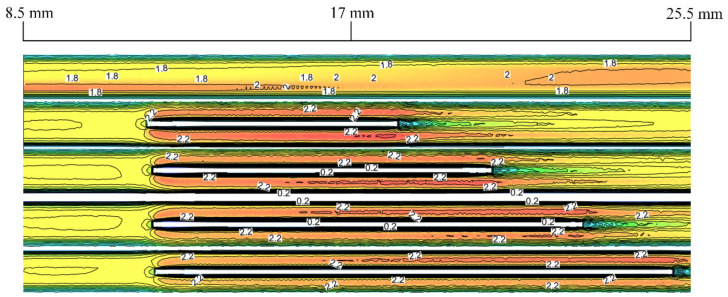
Velocity diagram of Cases 0–4 at the inlet velocity of 1.5 m/s.

**Figure 8 micromachines-13-01821-f008:**
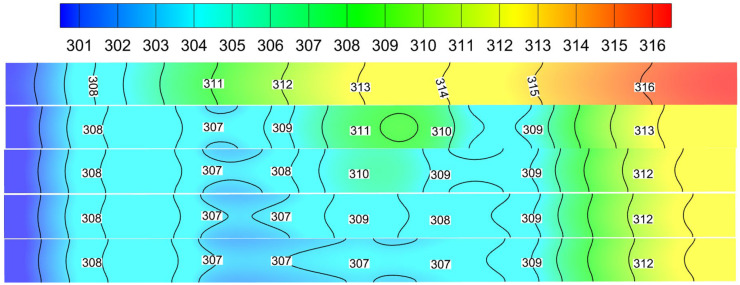
Temperature distribution diagram of Cases 0–4 at the inlet velocity of 1.5 m/s.

**Figure 9 micromachines-13-01821-f009:**
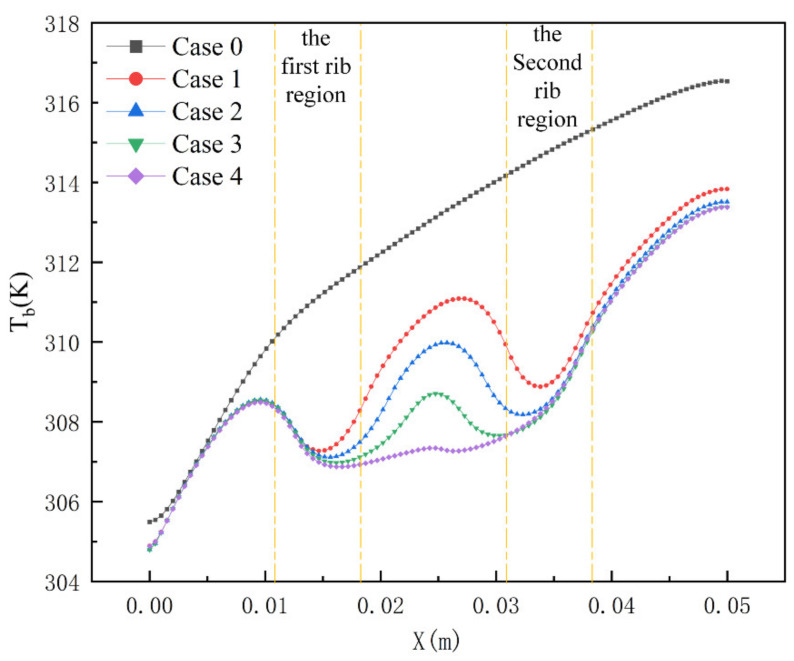
Temperature curve of Case 0–4 at the bottom centerline when the inlet velocity is 1.5 m/s.

**Figure 10 micromachines-13-01821-f010:**
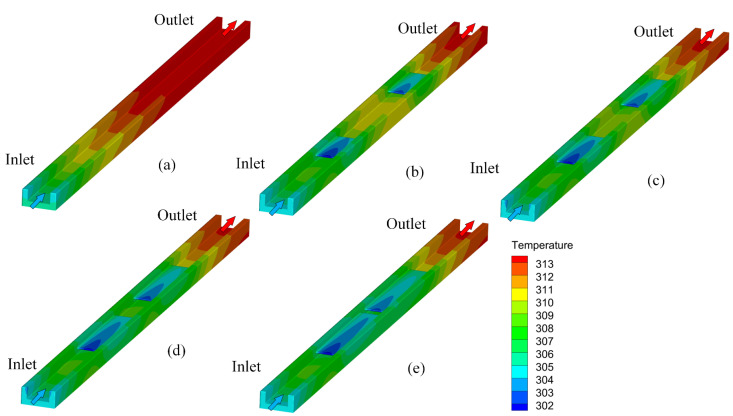
Temperature diagram of solid region at the inlet velocity of 1.5 m/s; (**a**) Case 0, (**b**) Case 1, (**c**) Case 2, (**d**) Case 3, and (**e**) Case 4.

**Figure 11 micromachines-13-01821-f011:**
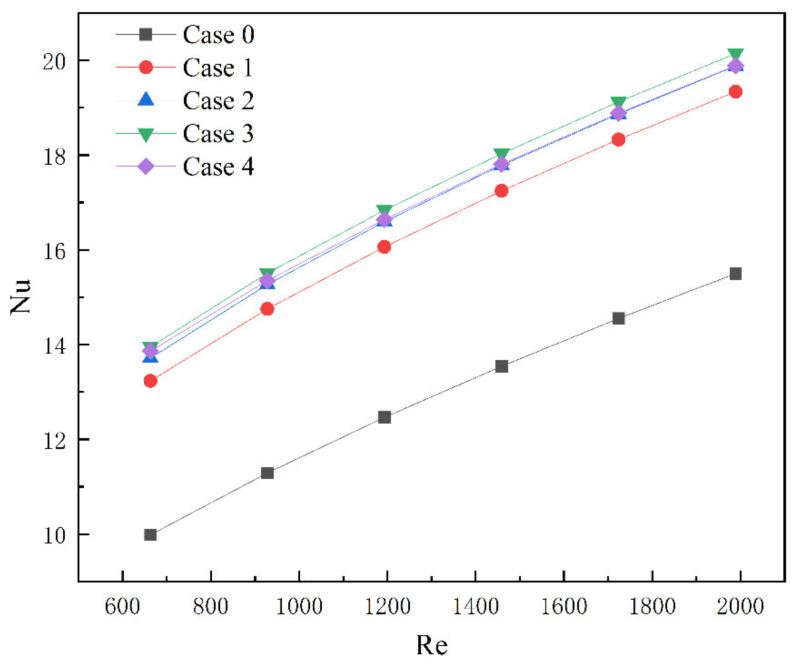
Nusselt number of Cases 0–4 at different Reynolds numbers.

**Figure 12 micromachines-13-01821-f012:**
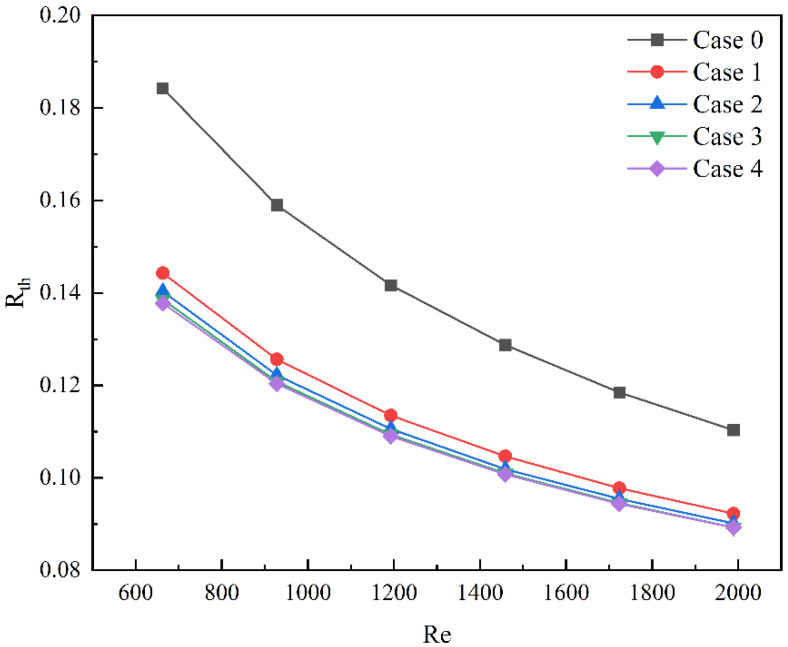
The thermal resistance of Case 0–4 at different Reynolds numbers.

**Figure 13 micromachines-13-01821-f013:**
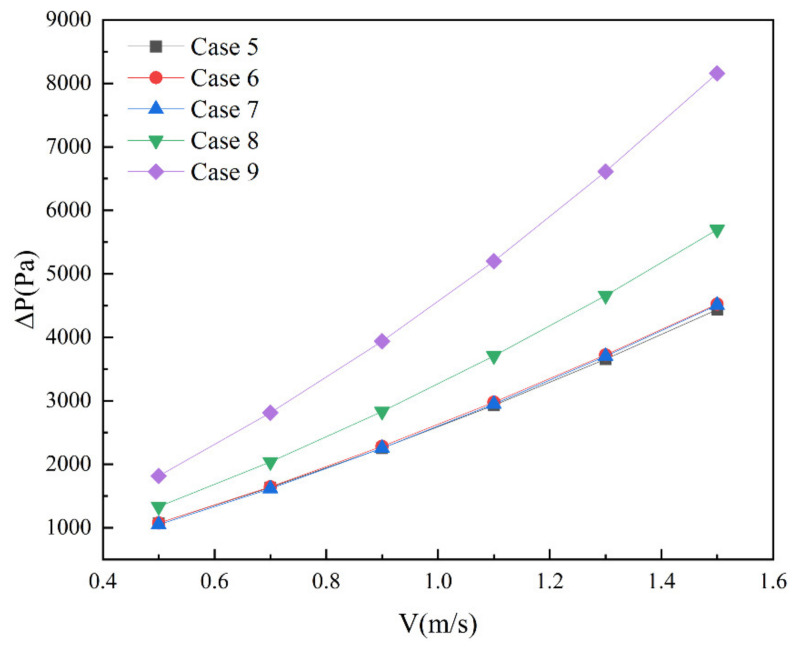
Pressure drop of Cases 0–4 at different inlet velocity.

**Figure 14 micromachines-13-01821-f014:**
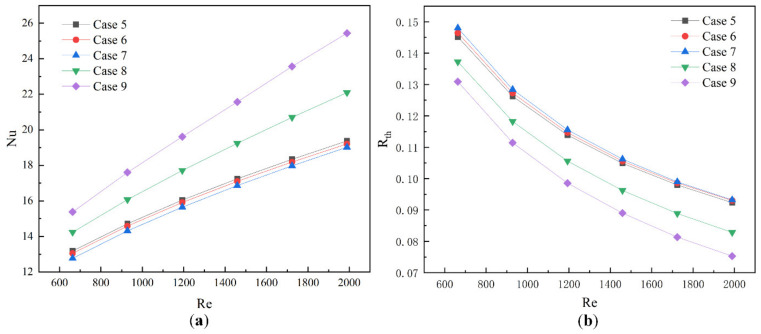
Data diagram of Case 5-9; (**a**) Nusselt number at different Reynolds numbers; (**b**) thermal resistance at different Reynolds numbers.

**Figure 15 micromachines-13-01821-f015:**
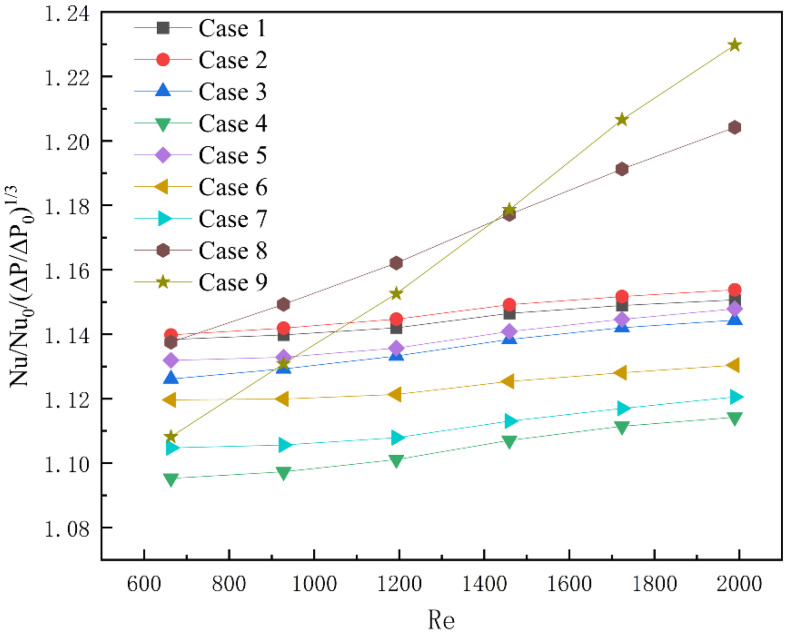
Comprehensive evaluation factors of different Cases at different Reynolds numbers.

**Table 1 micromachines-13-01821-t001:** Parameters of each case.

Model	L (mm)	θ (°)	d (mm)
Case 1	5.5	0	0.1
Case 2	7.5	0	0.1
Case 3	9.5	0	0.1
Case 4	11.5	0	0.1
Case 5	5.5	10	0.1
Case 6	5.5	15	0.1
Case 7	5.5	20	0.1
Case 8	5.5	0	0.2
Case 9	5.5	0	0.3

**Table 2 micromachines-13-01821-t002:** Grid Verification Data.

	Mesh I	Mesh II	Mesh III
Number of Grids	450,263	524,690	679,869
Nu	10.278	9.965	9.983
Difference	2.96%	0.18%	Baseline

## Data Availability

The data used to support the findings of this study are included within the article.

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
