# Peer review of "Numerical Study of Thermal Enhancement in a Single- and Double-Layer Microchannel Heat Sink with Different Ribs"

_micromachines, 2022, doi:10.3390/mi13111821_

Round 1

Reviewer 1 Report

Reviewer comments:

1. Explain the research gap observed and the novelty of the work clearly in the introduction.

2. Clarify the sentence “the substrate is 0.5mm” in section 2, page number 3.

3. In section 4.2, figure 11 shows the variation of the Nusselt number and thermal resistance

with the Reynolds number but the explanation is given for pressure drop distribution, please

correct it.

4. In section 4.3, explain the reasons for the mentioned effects.

5. In the title and abstract, the authors mentioned about double-layered heat sink but in the

physical model and the results section nothing showed about the double-layered heat sinks,

are the segments of heat sinks having the rib considered as a double layer? Please clarify.

6. In the literature, so much work is available on micro-channel heat sinks with advances in

geometry, fins, ribs, working fluid, jet impingement, secondary flows, etc., how the present

work is different and improved from existing works.

7. The heading of section 3.3 is “grid independence and model validation” but nothing is

mentioned about model validation, please correct it.

Author Response

请看附件

Reviewer 2 Report

I have several comments and concerns on the manuscript in the attached file. Based on these concerns it is recommended that the paper need major revision before considered for publication.

Round 2

Reviewer 1 Report

Novelty is missing

Reviewer 2 Report

The authors has made required corrections in the manuscript. So, it can be considered for publication in the journal.